∂ | **Open Peer Review** | Virology | Research Article

# Differential analysis of IBV-infected primary chicken embryonic fibroblasts and immortalized DF-1

Qingcheng Yang,[1] Huiling Gong,[1] Song Liu,[1] Siyu Huang,[1] Wenjun Yan,[1] Kailu Wang,[1] Hao Li,[1] Chang-Wei Lei,[1] Hong-Ning Wang,[1] Xin Yang[1]

**ABSTRACT**  Infectious bronchitis virus (IBV), the causative agent of infectious bronchitis, is responsible for major economic losses in the poultry industry worldwide. While IBVs can usually be passaged in primary chicken embryonic fibroblasts (CEFs), most of the wild ones cannot adapt to passaged cell lines. In this study, the wild strain CK/CH/MY/2020 was used to infect primary CEF and immortalize DF-1 CEF cells. Results indicated that IBV was able to cause lesions and pass onto CEF, but not DF-1. Indeed, the virus could enter DF-1 cells and synthesize the associated structural gene but could not assemble into complete viral particles for release. Furthermore, transcriptome sequencing analysis showed significant differences in gene expression between CEF and DF-1 cells after viral infection, although the corresponding antiviral responses could be activated in both cell types. The biggest difference was in terms of the amino acid biosynthesis pathway and the cytokine receptor interaction pathway, which were significantly and specifically activated in CEF. This could actually explain why intact viruses can be assembled but not in DF-1. In addition, *SLBP* and *P2RX7* affect the replication of IBV's structural genes to some extent. Overall, IBV can enter CEF and DF-1 cells, but the complex intracellular cytokine interactions affect the assembly and release of viral particles. The insight will be useful for the study of IBV through *in vitro* transmission and pathogenesis.

**IMPORTANCE** Infectious bronchitis virus (IBV) is responsible for high morbidity and mortality as well as substantial economic losses worldwide. Transcriptome sequencing of IBV-infected chicken embryonic fibroblast and DF-1 cells revealed that the virus elicits antiviral immunity in cells after viral infection, but IBV cannot activate DF-1 cells to produce sufficient amounts of viral structures to assemble into complete virions, which may be caused by the interactions between cytokines. The study of IBV cellular adaptations is important for vaccine development and investigation of the pathogenesis of IBV.

**KEYWORDS**  infectious bronchitis coronavirus (IBV), cell orientation, transcriptome sequencing, immunization pathway

nfectious bronchitis virus (IBV), the causative agent of infectious bronchitis (IB), has been responsible for major economic losses in the poultry industry worldwide ever since it was first reported in 1931 (1). IBV belongs to the coronavirus family (*Gammacoronavirus* genus of the Coronaviridae family) and is a positive-stranded RNA virus with a genome of approximately 27–32 kb. *In vitro* culture of viruses is critical for studying the mechanism of viral infections. Once a virus infects a cell, the viral genome is released, and, with the help of the cellular machinery to synthesize the structural proteins of viruses, the complete viral particle is assembled and released from the cell for the next round of infection. The IBV virus particle consists of four structural protein packages,

Address correspondence to Xin Yang, yangxin0822@163.com.

The authors declare no conflict of interest.

namely, the nucleocapsid (N) protein, the coronavirus spines (S), the membrane (M) protein, and the envelope (E) protein. Of these, the N and M proteins play an important role in the assembly of the viral particles (2–4).

Although IBV is widely distributed in the epithelium of host tissues, most strains have a highly restricted cytophilic nature. Indeed, they can only replicate in primary chicken kidney cells, primary chicken embryo cells, and chicken embryos, while some isolates only infect chicken embryos. Traditionally, primary chicken embryonic fibroblasts (CEFs) have been used in virology and vaccine production, although a major disadvantage remains the fluctuation of virus titers between lots. Thus, there are transformed cell lines for vaccine production as they provide an unlimited supply of identical cells (5). DF-1 is one such continuous cell line of CEF (6). The cells are free of endogenous sequences related to avian sarcoma and leukosis viruses and have normal fibroblastic morphology.

The only IBV Beaudette strain that is adapted to multiple passaged cell lines was also initially available only *in vitro* using chicken embryos. However, they were passed only three times before being transferred to Vero cells for 65 consecutive passages by Fang et al. The Beaudette-adapted strain was finally obtained to be stably inherited in Vero cells (7–9). With variations between IBV strains and an increase in the number of genotypes and serotypes, some studies have found the potential of a few isolates to adapt to passaged cell lines. For instance, the recombinant strain isolated by Huang et al. (10) was able to cause lesions not only in primary CEK cells but also in DF-1 cells, with N gene upregulation and N protein expression detected except during successive passages. This highlights the importance of studying the biological properties of new strains in addition to ongoing epidemiological investigations and molecular surveillance of IBV. This is because the susceptibility of IBV to passaged cell lines limits both vaccine development and research on the mechanism of viral infection of cells (11–13).

In this study, strain CK/CH/MY/2020, which was previously isolated in the authors' laboratory (14), was used in order to observe differences between the infection of DF-1 and CEF cells. After several cell passages, the transcriptomes of DF-1 and CEF cells were compared to investigate differences in the adaptability of the wild strain to the two types of cells. The results are expected to provide a reference for scholars to conduct *in vitro* cell passages when studying IBV strains.

## RESULTS

### IBV can be transmitted in CEF but not in DF-1

IBV was transmitted blindly across CEF and DF-1 cells, and it was found that, after infection, the former showed significant lesions, while DF-1 grew well (Fig. 1a). PCR amplification of cell supernatants from three generations of passages subsequently revealed that the target bands were present in all CEF groups, while in the case of DF-1, they were present only in the first generation (Fig. 1b). In addition, the IBV structural genes were amplified, with the results showing that they were present only in the CEF supernatant (Fig. 1c). The expression of the virus's four structural genes was then determined by isolating cellular RNA at different time periods. CK/CH/MY/2020 highly expressed the N gene after infecting DF-1, with a peak in expression noted at 24 h. The expression of the N gene then decreased rapidly until at 48-h post-infection where it was approximately similar to that at 8-h post-infection. Overall, although trends in the expression of the E and N genes were similar, the former was much lower. As far as the CEF cells were concerned, the expression of the N gene also peaked 24 h after CK/CH/MY/2020 infection. This was followed by slight fluctuations in expression that nevertheless remained at a high level until the end of the observation. The expression of the M gene was similar to that of the N gene at 36-h post-infection, but there was a subsequent increase in expression levels between 36- and 48-h post-infection until a peak was reached at 48 h. Compared with the M and N genes, the expression of S and E genes was lower in CEF, with that of the E gene peaking at 24-h post-infection and stabilizing within a certain range after a small regression. In contrast, expression of the S

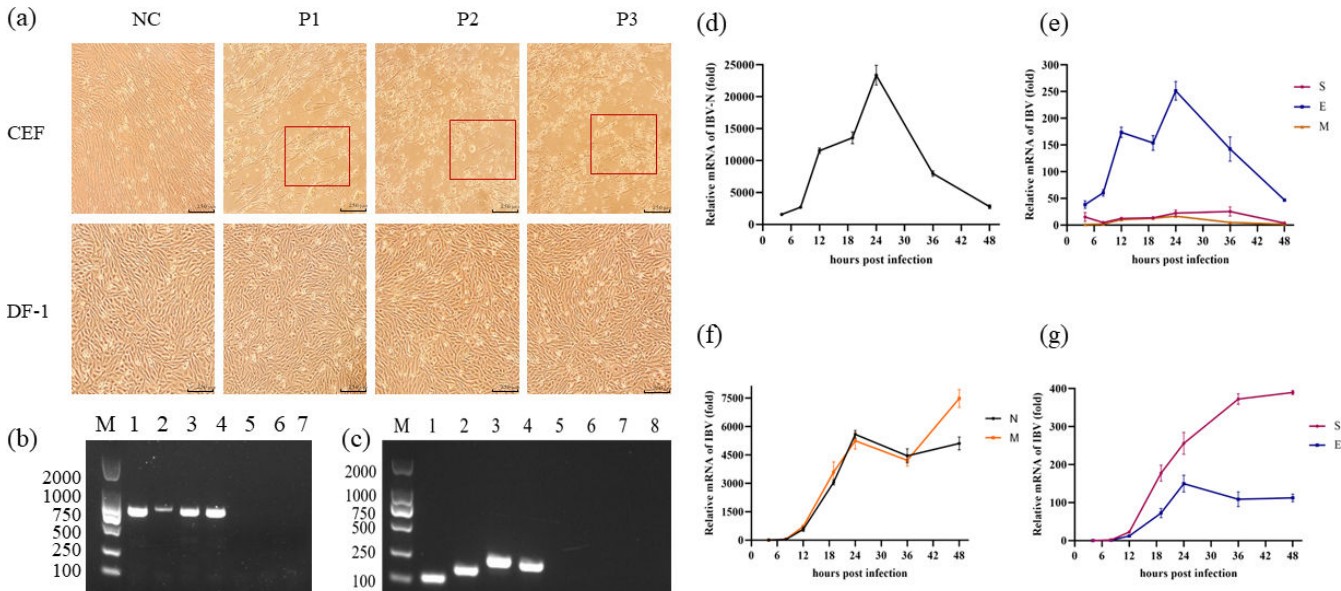

**FIG 1** Virus transmission in CEF and DF-1 and amplification of structural genes. (a) Viral transmission on CEF and DF-1. (b) PCR amplification junction results for three generations of cell supernatants. M:DL2000: 1–3: blind transmission of the virus in CEF for generations 1–3; 4–6: blind transmission of the virus in DF-1 for generations; 7: negative control. (c) PCR results for viral structures in the supernatant of blind passaged third-generation cells. M:DL2000: 1–4: S, E, M, and N genes in the supernatant of CEF; 5–8: S, E, M, and N genes in the supernatant of DF-1. (d and e) Structural genetic changes after viral infection with DF-1. (f and g) Structural genetic changes after viral infection with CEF.

gene increased with increasing infection time, with a high level of expression reached at 48-h post-infection (Fig. 1d to g).

The presence of IBV's N protein was detected in both types of cells after viral infection using mice anti-IBV N protein as the primary antibody (Fig. 2a). Surprisingly, the green fluorescence was observed in about one-third of the CEF group, while for the DF-1 group, there were no lesions despite the green fluorescence.

Sections of the CEF group clearly showed a number of vesicular structures containing fully assembled virus particles in the cytoplasm. Complete viral particles that had been expelled could also be observed outside the cells. In contrast, the cell morphology was intact in the DF-1 group, with only a small amount of autophagosome production (Fig. 2b).

## CEF and DF-1 have widely divergent gene expression patterns

CEFs are primary CEFs that cannot be continuously passaged *in vitro*, while DF-1 is a transmissible chicken fibroblast cell line derived from CEF cells. In this study, the transcriptome of both cell types was analyzed and compared, and as many as 6,398 differentially expressed genes were identified. The volcano plot, presented in Fig. 3a, shows that 2,616 genes were actually upregulated based on the threshold of |log2 fold change (FC)| > 2, while 1,135 were downregulated. Gene ontology (GO) enrichment (Fig. 3b) analysis further showed that the differentially expressed genes were mainly concentrated in the extracellular region (extracellular matrix, enzyme matrix, and the enzyme matrix) and the functional regions (extracellular region part, extracellular matrix, enzyme inhibitor activity, metallopeptidase activity, and DNA replication and regulation of growth). Kyoto Encyclopedia of Genes and Genomes (KEGG) analysis also (Fig. 3c) revealed that the differential genes were mainly involved in the Fanconi anemia pathway, cell cycle, neuroactive ligand–receptor interaction, and calcium signaling pathway. Overall, being immortalized fibroblasts, it is not surprising that many genes related to cell cycle, cell adhesion, and migration are altered in DF-1 as molecular mechanisms that

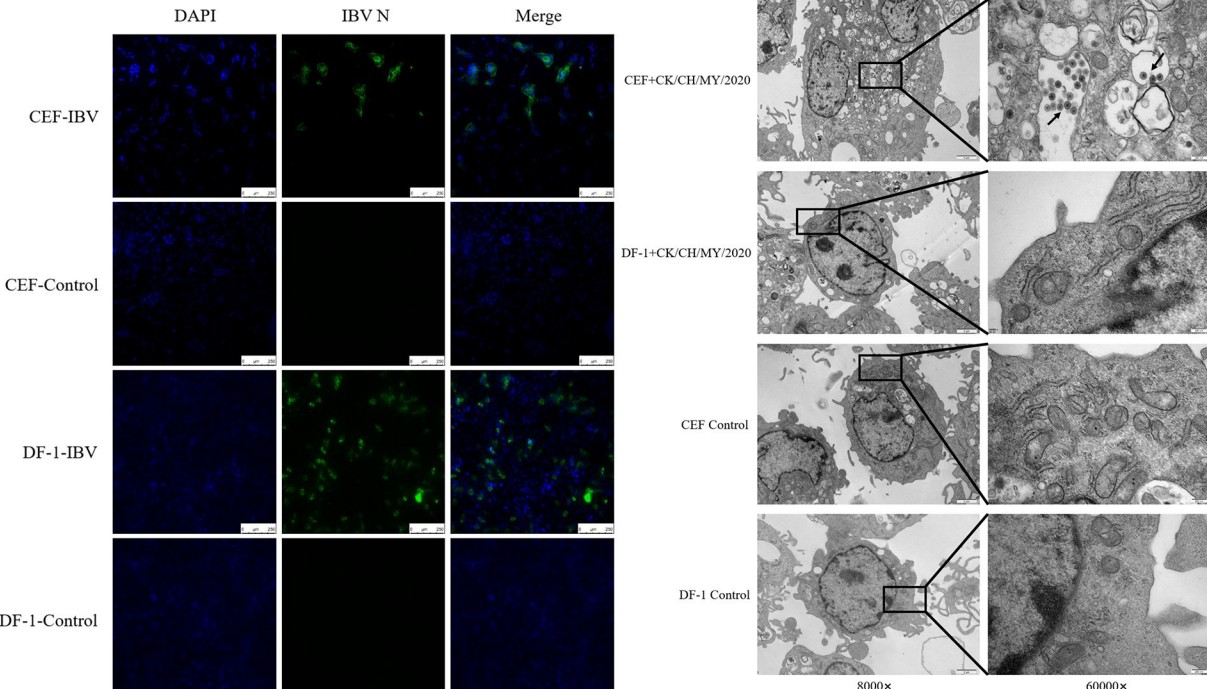

**FIG 2** Indirect immunofluorescence assay (IFA) and electron micrographs of CEF and DF-1 cells after 24 h of virus infection. (a) Identification of IBV-infected cells by IFA. Anti-IBV N monoclonal antibody was used for the detection of IFA. (b) Electron microscope image.

enhance cell cycle progression, inhibit cell death pathways, alter cell morphogenesis, and accelerate molecular transport capacity contribute to their proliferation *in vitro*.

## Transcriptome analysis before and after CEF infection

There were 1,053 significantly upregulated genes and 489 significantly downregulated ones after CEF infection by the virus, with the results presented in the volcano plot shown in Fig. 3d. The significantly upregulated genes included *USP41*, *ZC3HAV1*, *TLR3*, and *STAT1*, just to name a few, and they were mostly involved in antiviral effects, immune induction, cell migration, and tumorigenesis. Similarly, the significantly downregulated genes included *CHAC1*, *SLC7A11*, *PYCR1*, *PSPH*, and *DDIT4*, among others, with these genes being involved in iron death, cancer cell expansion, and apoptosis (15–19). GO and KEGG analyses of all significantly altered genes are also shown in Fig. 3e and f. In this case, GO analysis indicated that these differential genes were mainly enriched in chemokine activity (chemokine), chemokine receptor binding (chemokine receptor binding), and cytokine receptor, while KEGG enrichment analysis showed that the differential genes were mainly enriched in the molecular functions (MFs) of nucleotide oligomerization domain (NOD)-like receptor signaling, influenza A virus, simplex virus herpesvirus type 1 infection, cytokine receptor interactions, and Toll-like receptor signaling pathway as well as other pathways. Altogether, the results showed that CEF triggered strong antiviral responses and induced cellular immune responses after viral infection.

## Transcriptome analysis before and after DF-1 infection

For this cell type, a small number of genes were found to have significantly altered expressions based on transcriptome assays. More specifically, the significantly upregulated genes (Fig. 3g) included *IFIT5*, *CMPK2*, *MX1*, *IFI6*, and *LY6E*, all of which are associated with important roles during antiviral processes (20–22). Furthermore, based on GO analysis (Fig. 3h), the differential genes were found to be mainly involved in GTPase

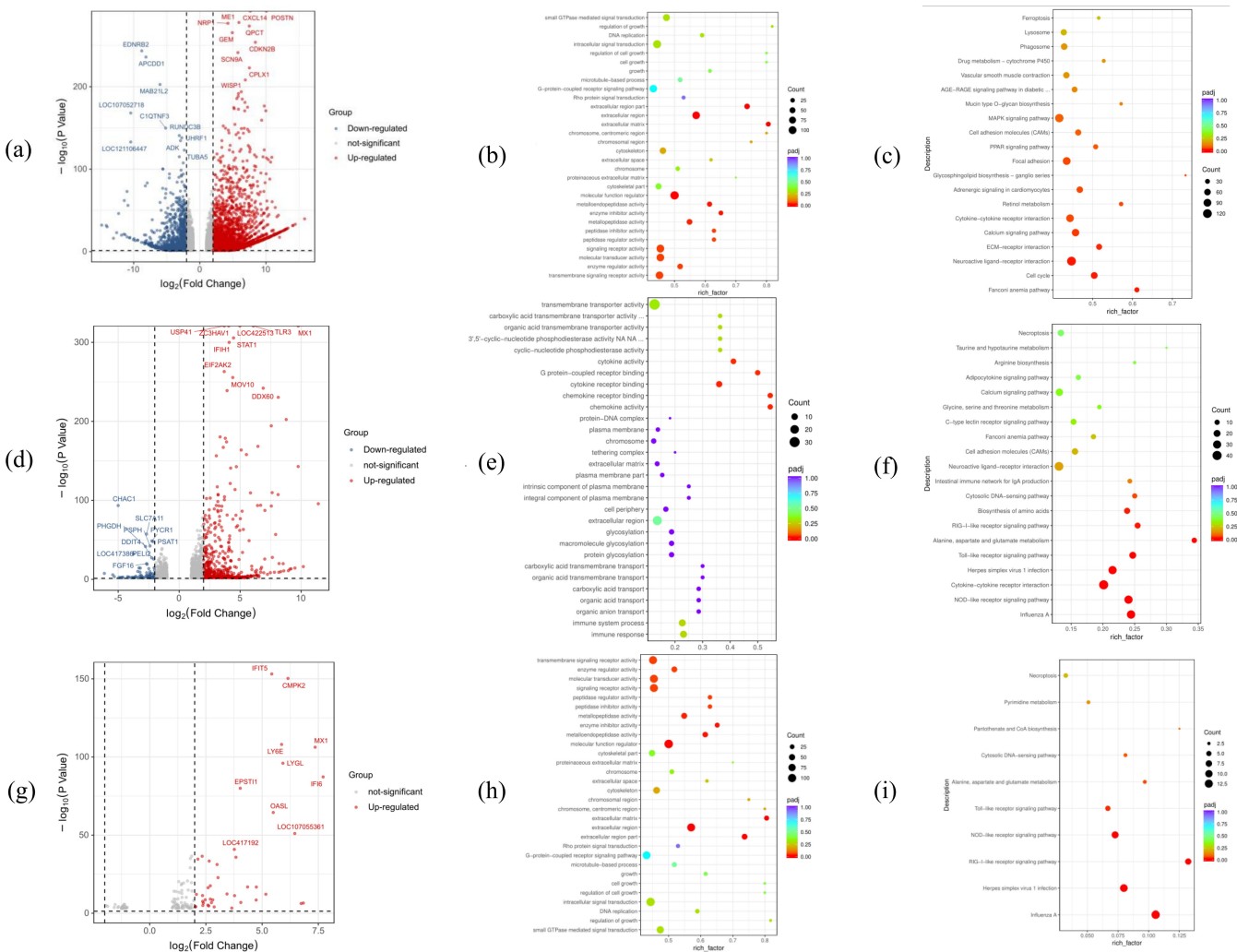

**FIG 3** Results of transcriptome analysis. (a–c) Comparison of CEF and DF-1 transcriptomes without virus infection: (a) volcano map; (b) GO enrichment analysis; and (c) KEGG-enriched pathways. (d–f) Comparison of CEF's transcriptome before and after viral infection: (d) volcano map; (e) GO enrichment analysis; and (f) KEGG-enriched pathways. (g–i) Comparison of DF-1's transcriptome before and after viral infection: (g) volcano map, (h) GO enrichment analysis, and (i) KEGG-enriched pathways.

activity, NAD+ ADP-ribosyltransferase activity, transferase activity, transferring pentosyl groups, and other catalytic reactions. Similarly, KEGG enrichment analysis, shown in Fig. 3i, indicated that after infection, the differential genes in DF-1 cells were mainly enriched in influenza A, herpes simplex virus 1 infection, retinoic acid-inducible gene I (RIG-I)-like receptor signaling pathway, NOD-like receptor signaling pathway, Toll-like receptor signaling pathway, alanine, aspartate and glutamate metabolism, and cytosolic DNA-sensing pathway, which were also significantly enriched after CEF infection. Two pathways, in particular, namely, cytokine–cytokine receptor interaction and biosynthesis of amino acids, were not significantly enriched in DF-1 after virus infection in CEF compared to DF-1. Hence, even though CK/CH/MY/2020 may not be able to cause cytopathic lesions, it can still induce intracellular antiviral responses to some extent, while for CEF, it is still able to infect by activating the amino acid biosynthesis pathway to synthesize proteins required by the virus.

The activation of some inflammatory factors and interferon-stimulated genes indicates the release of the viral genome into the cytoplasm, thereby further suggesting that wild IBV can enter DF-1 cells. However, the likelihood that DF-1 could not get the virus out of the cells due to a lack of IBV receptors was ruled out.

## Analysis of differential genes between CEF and DF-1 after viral infection

For both cell types, genes with significantly changed expression after viral infection were compared. It was found that, out of 1,542 such genes in CEF cells, 91 overlapped with those of DF-1 cells after infection, while the remaining 1,451 ones were specific to CEF. These two sets of genes were then compared separately.

Eighty-five of the 91 genes were upregulated and included *IFIT5*, *TLR3*, *IFI6*, *MX1*, and *CMPK2*, most of which are immune-related (16, 20, 22–25). Figure 4b shows that, although the trends in gene expression were similar for CEF and DF-1 after viral infection, there were more variations in the case of CEF, hence indicating a stronger immune response. GO enrichment analysis (Fig. 4c) then showed that the genes were mainly enriched in NAD+ ADP ribosyltransferase activity [GTPase activity, transferase activity, and transfer of pentosan moieties (hydrolase activity, acting on acid and other molecular biological functions)]. Moreover, KEGG enrichment analysis, shown in Fig. 4d, indicated that these 91 genes were mainly concentrated in pathways, such as influenza A virus, herpes simplex virus type 1 infection, RIG-I-like receptor signaling pathway, and NOD-like

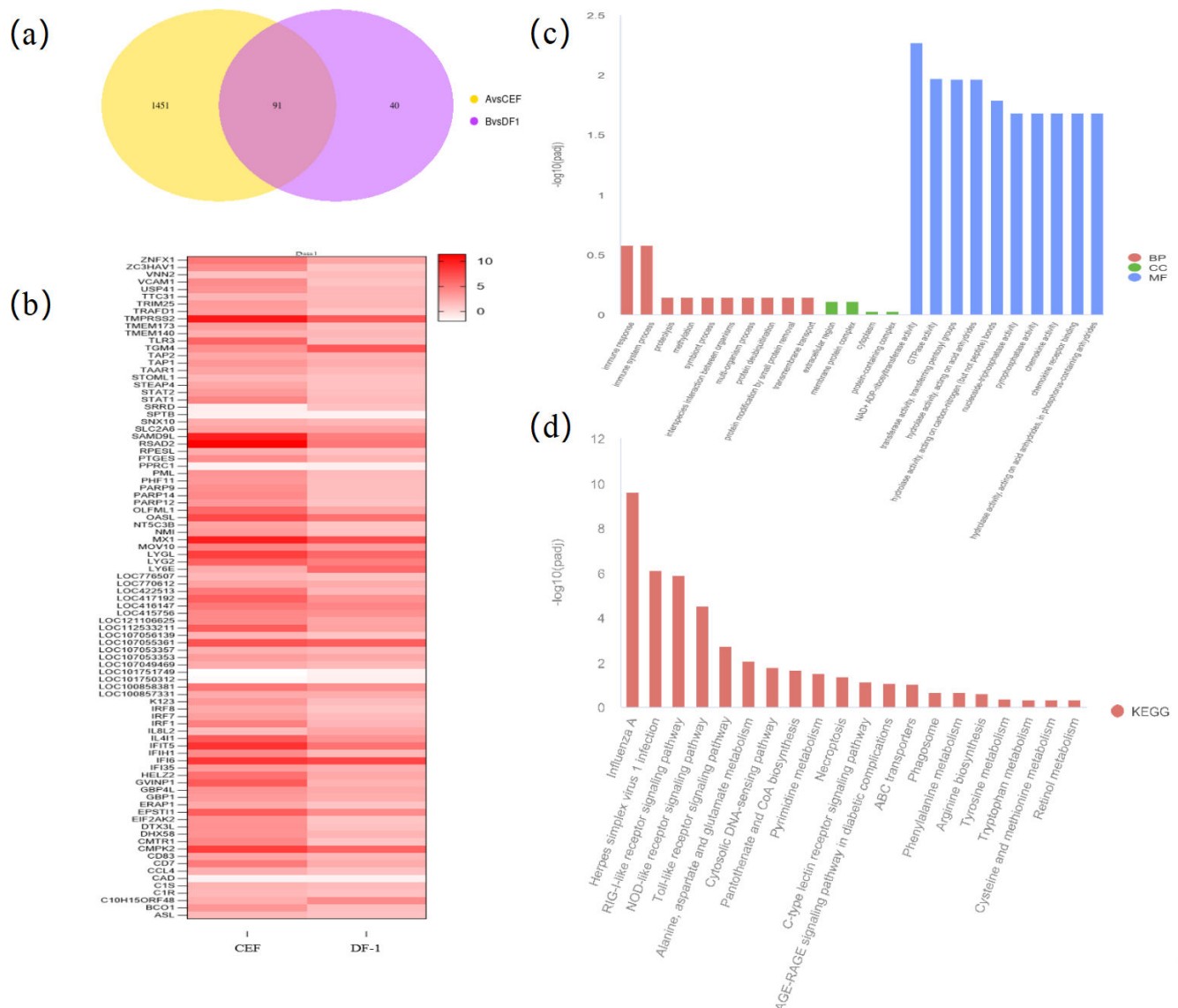

**FIG 4** Genetic correlation analysis of common changes in CEF and DF-1 after viral infection. (a) Coexpression of Venn diagram. (b) Gene clustering heat map. (c) GO enrichment analysis. (d) KEGG enrichment analysis.

receptor signaling pathway, with these results being consistent with the KEGG analysis obtained after infection of DF-1. The findings indicate that these genes and pathways mainly play a role against diseases.

Among the 1,451 genes whose expressions were specifically altered in CEF after infection, some were involved in immune-related functions, while others, such as the significantly upregulated *SLBP*, *P2RX7*, *GGT5*, and *RNF213*, could promote viral particle replication or enhance the effect of infection. On the other hand, the downregulated genes were mainly involved in cell adhesion, ion channels, immune regulation, and cell death, among others. Interestingly, 37 of these 1,451 genes were also considered potentially novel genes.

Figure 5a shows the GO enrichment analysis of these 1,451 genes that were enriched in all three GO terms, especially in biological processes (BPs) such as cell population proliferation, immune effect over, positive regulation of metabolic processes, cellular modification of amino acid metabolic processes, and leukocyte activation, among others. Figure 5b shows the involvement of individual genes in specific processes of BP. For instance, 22 genes, including *IFNG*, *WNT2B*, *GATA2*, and *NFIB*, were involved in cell population proliferation, while 37 genes, including *IFNW1*, *IFNG*, *MYD88*, and *TBXT*, were involved in positive regulation of metabolic processes. Finally, *IFNW1*, *GATA3*, *MYD88*, *PRDX1*, and others were involved in several biological processes.

Figure 5c shows the KEGG enrichment analysis that indicated that the genes were enriched in cytokine receptor interactions, amino acid biosynthesis, herpes simplex virus type 1 sense, alanine, aspartate, and glutamate metabolism, NOD-like receptor signaling pathway, and Toll-like receptor signaling pathway. Figure 5d further shows the attribution of each gene to the pathway. Thus, 34 genes, including *IFNW1*, *IFNG*, *IL18R1*, *CCL1*, and *CCL19*, were involved in the cytokine receptor interaction pathway, while 18 genes, including *P2RX7*, *GBP*, *RIPK2*, and *PANX1*, were involved in the NOD-like receptor signaling pathway.

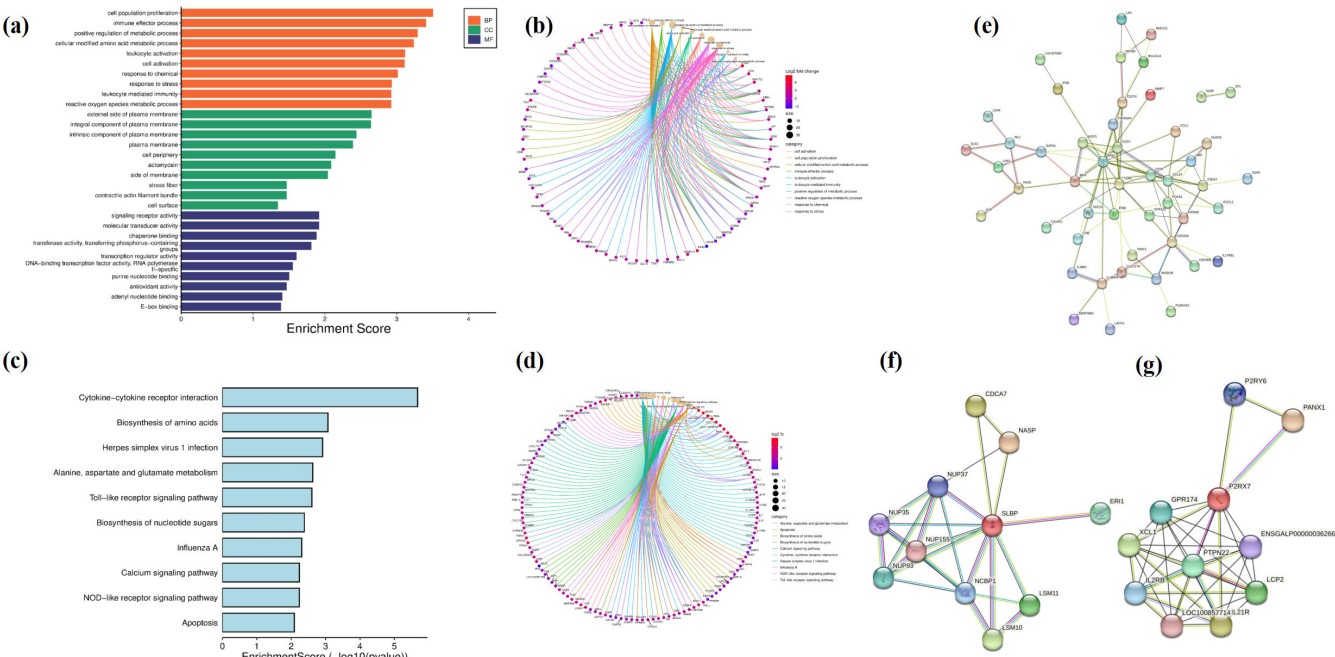

FIG 5 Enrichment analysis of 1,451 genes with changed expression specifically in CEF after infection. (a) GO enrichment analysis. (b) Diagram showing the relationship between genes in BPs. (c) KEGG enrichment analysis. (d) Diagram showing the relationship between genes in each pathway. (e) Protein interactions between 50 significantly upregulated genes. (f) *SLBP* protein interplay map. (g) *P2RX7* protein interaction diagram.

## *SLBP* and *P2RX7* can partially promote the replication of IBV's structural genes in DF-1

The *SLBP* gene is mainly involved in RNA binding, while the *P2RX7* gene is mainly associated with the NOD-like receptor signaling pathway. Although *SLBP* genes were not in the key KEGG pathway analysis, they were included as important because of their high differential expression as well as their pro-infection role as demonstrated in previous viral studies (26). Protein interaction analysis was also performed using the STRING online website for 50 genes that were significantly upregulated in DF-1 and CEF, with the resulting interaction network shown in Fig. 5e in which the SLBP and P2RX7 proteins are marked with black circles. The *SLBP* gene is associated with only the *ZP1* gene, while *P2RX7* is associated with *CCL19*, *NLRC5*, *GDPD*, and *GBP*. In fact, *P2RX7*, as the key network of nodes, is mainly associated with the NOD-like receptor signaling pathway (Fig. 6). In this pathway, P2RX7 mainly acts as a cation channel on the cell membrane to mediate the entry of $K^+$ into the cell.

The protein interaction network map using the *SLBP* and *P2RX7* genes alone, as shown in Fig. 5f and g, shows that *SLBP* is mainly associated with histone mRNA metabolic processes, histone binding to mRNA stem loop, and nuclear membrane nuclear pore composition, and, as such, it is mainly involved in ribonucleic acid transport-related pathways. *P2RX7*, on the other hand, is mainly associated with the positive regulation of interleukin-1α production.

*P2RX7* is also related to the positive regulation of α-β-T-cell proliferation, the positive regulation of T-cell-mediated cytotoxicity, and the cytokine receptor interaction pathway.

DF-1 cells overexpressing the *SLBP* and *P2RX7* genes were named SLBP-DF-1 and P2RX7-DF-1, respectively. Neither the experimental group nor the control group showed a cytopathic effect. The expression of the E gene was lower in both overexpressed DF-1 cells compared with the normal ones. Regarding the M gene, its expression was highest in SLBP-DF-1 cells and peaked at 36 h, but it was low in normal DF-1 cells and decreased after peaking at 24 h. At the same time, the expression was almost absent in P2RX7-DF-1 cells. The trend for N gene expression was consistent in all three cells, peaking at 24 h. More specifically, the expression was similar in SLBP-DF-1 and normal DF-1 cells but low in P2RX7-DF-1 cells. Finally, the expression of the S gene peaked in all three cell types at 36-h post-infection, with the highest expression observed in normal DF-1 cells. However,

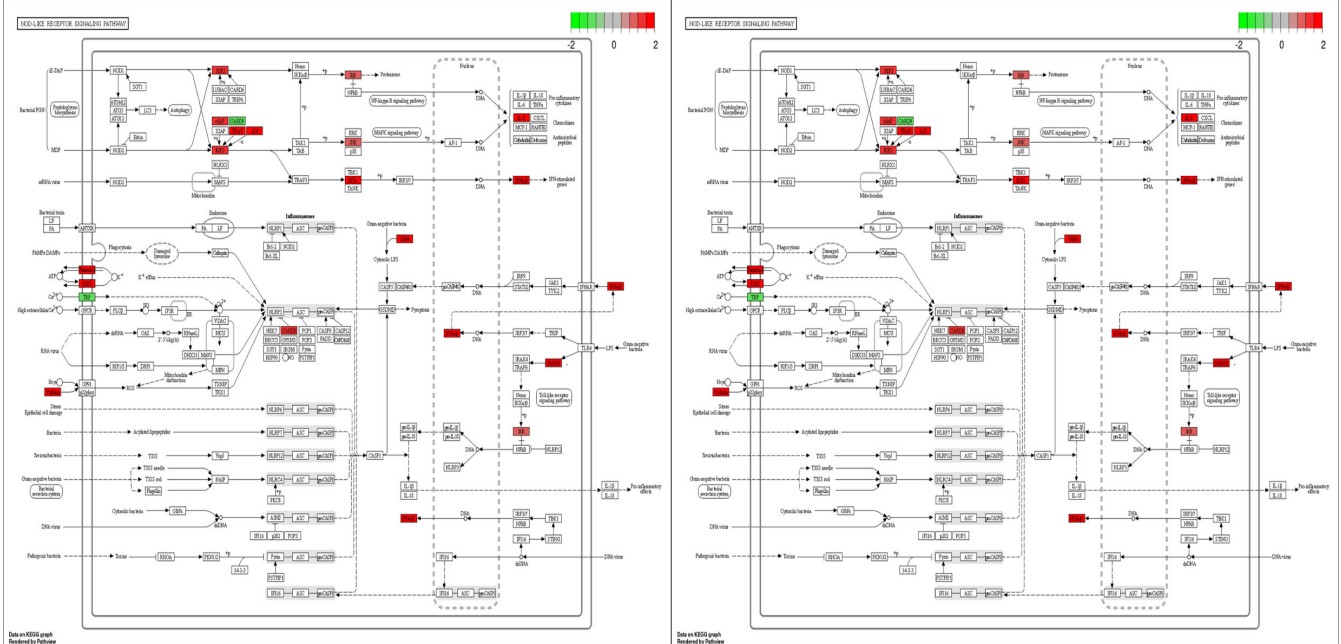

**FIG 6** NOD-like receptor signaling pathway diagram (red for upward adjustment; green for downward adjustment).

the low expression occurred in SLBP-DF-1 cells, with P2RX7-DF-1 cells displaying the lowest (Fig. 7).

The *P2RX7* gene reduces the replication of four structural genes after overexpression. *SLBP* can promote the value of CK/CH/MY/2020's M genes to some extent after overexpression, but the other structural genes do not undergo corresponding changes. In addition, no obvious large-scale intact viral particles could be observed in the electron micrograph (Fig. 7f). Although DF-1 originates from CEF cells after immortalization, the biochemical processes, cellular metabolism, and other factors between the two differ significantly. The complete life cycle of a virus requires the precise cooperation of many proteins in the host cell.

The reasons for the inability of CK/CH/MY/2020 to replicate into intact viruses in DF-1 could be extremely complex, thereby highlighting the need for subsequent experiments.

## DISCUSSION

All wild-type IBV strains, except for the Beaudette strain, are capable of continuous passaging across avian immortalized cell lines such as DF-1, HD11, and chicken liver cancer cells (LMH), but they are not necessarily adaptable to the passaging process, which greatly hinders the study of the mechanism of IBV invasion as well as applications for vaccine production (9). Many authors have therefore investigated the factors affecting IBV cytophagy and found that one of the determinants was the viral S gene, with the S1 protein being responsible for cell recognition and the S2 protein being responsible for facilitating the fusion of the cell membrane with the capsid to facilitate virus entry into the cell. Cofactors such as cell receptors are also present in the host cell to help the virus enter the cell. However, entry into the cell is only the first step of invasion; successful replication, transcription, translation, assembly, and release in the cell are subsequently required to achieve virus proliferation (12). It has been shown that when a virus invades a cell, in addition to infectious and non-infectious forms, it also exhibits an intermediate state of "semi-infection" during which the virus is able to replicate and transcribe viral genes in the cell as well as translate some or even all of the structural proteins without being able to successfully assemble complete viruses. This situation is referred to as viral stuttering or abortive infection (24, 27). For example, canarypox virus and chicken pox virus, both of which can infect avian species normally, will display a

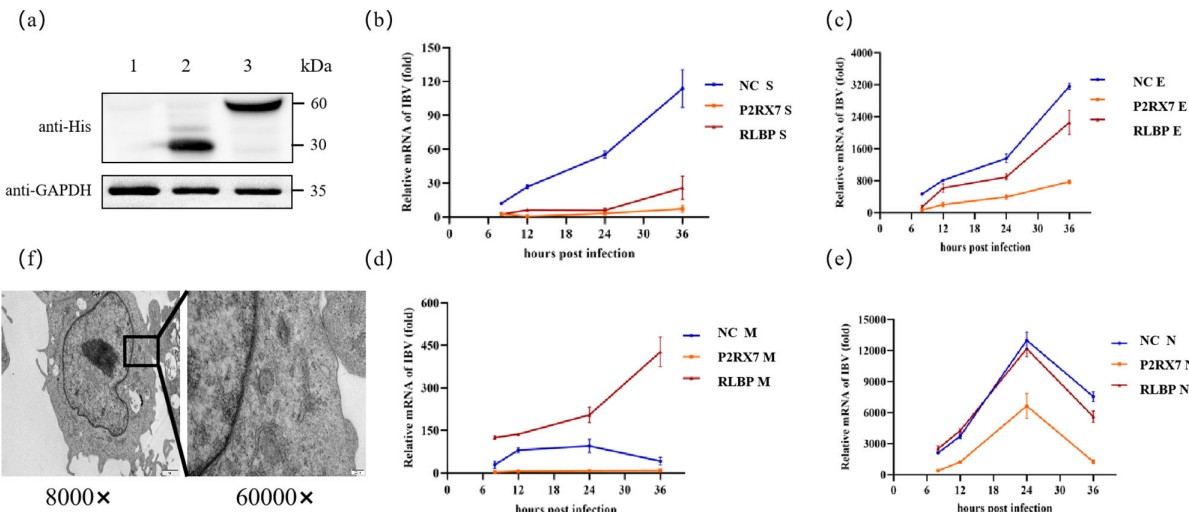

**FIG 7** Protein expression of SLBP and P2RX7 and effects on viral structural protein expression. (a) Western blot measurement of SLBP and P2RX7 protein expression (1: blank control; 2: SLBP protein; 3: P2RX7 protein). Anti-His monoclonal antibody was used for the detection of IFA. (b–e) Changes in the RNA expression of structural genes after CK/CH/MY/2020 infection with SLBP-DF-1, P2RX7-DF-1, and DF-1 (blue: blank control; red: SLBP; orange: P2RX7): (b) changes in the S gene; (c) changes in the E gene; (d) changes in the M gene; and (e) changes in the N gene. (f) Electron micrograph showing CK/CH/MY/2020 infection of SLBP-DF-1 after 24 h.

similar ability to infect mammals by replicating and expressing genes and synthesizing complete protein particles during the early stages of infection, but eventually, the viruses are unable to reproduce normally.

In this study, the isolate CK/CH/MY/2020 was used to infect DF-1 and CEF cells, and it was found that the strain could adapt to the latter, causing cytopathic effects during successive passages. On the other hand, the strain did not cause cytopathic effects in DF-1 cells that remained unaffected after multiple blind passages, although reverse transcription quantitative PCR (RT-qPCR) of RNA extracted from infected cells revealed that the viral structural genes could still proliferate in the cells during the early stages of infection. This was especially the case for the N gene that was expressed at a very high level 24 h after infection. Subsequent observation of post-infected cells with IFA showed that the N protein could also be expressed normally in both cells. However, the expression of other structural genes could not be demonstrated due to the absence of corresponding specific antibodies. These results suggest that CK/CH/MY/2020 is able not only to enter DF-1 cells but also to use the host environment for self-gene replication and proliferation and for necessary protein expression. To further investigate why CK/CH/MY/2020 could not pass across DF-1 cells, the assembly of viral particles after treatment was observed by filming the cells 24 h after infection. In this case, the results showed that, in CEF cells, CK/CH/MY/2020 was able to form complete viral particles, some of which were released extracellularly. Therefore, it is likely that some wild IBV strains cannot proliferate on passaged cells due to their inability to assemble complete daughter viruses, resulting in a "stutter infection." However, the reason for this phenomenon remains unclear and thus would need to be further investigated.

Although CK/CH/MY/2020 was unable to induce lesions in DF-1, transcriptome analysis revealed that the expression of 131 genes was significantly altered in this cell type after infection with the virus, and these were mainly associated with antiviral responses such as inflammatory factor generation, immune response, and T-cell activation. However, in CEF, CK/CH/MY/2020 was able to infect successive generations, with transcriptome comparison revealing significant changes in the expression of 1,542 (including 37 unknown or potentially new genes) genes after viral infection. Of these, 91 were identical to those identified in DF-1, and they were mainly focused on immune-related pathways. To further compare differences in CK/CH/MY/2020 infection between CEF and DF-1, the remaining 1,451 significantly altered genes that did not overlap were analyzed. These genes were involved in cell transfer, cytokine release, the interferon pathway, the induction of immunity, and cell death, with most of these functions being associated with antiviral effects. These results are supported by studies (28, 29) where viral infestation inevitably caused organisms to resist, thereby eliciting an immune response as well as inducing apoptosis.

In comparing the immune pathways in which the differential gene sets of CEF and DF-1 were located, it was found that the pathways were consistent in both CEF and DF-1, with only two pathways, namely, the cytokine–cytokine receptor interaction and the biosynthesis of amino acids, being unique to CEF. These results indicate that CEF underwent virus-associated amino acid and protein biosynthesis after infection by the virus.

Among the significantly upregulated genes, two that were of interest were *SLBP* and *P2RX7*. Previous studies have shown that SLBP binds to replication-dependent (RD) histone 3′-end stem loop structures to aid in the maturation and translation of histone mRNA. However, human cytomegalovirus infestation prevents the synthetic accumulation of cellular S-phase RD histones and uses these on the bound SLBP to support viral replication. Production of infectious viral particles (26) may be useful in our understanding of the causes of viral tonoplast infection. P2RX7 may act as a danger sensor, activated by ATP and other nucleotides (e.g., NAD), to signal inflammatory programs to cells and play a role in immune regulation and tolerance. Several studies have suggested that *P2RX7* may be involved in the intensification of inflammatory responses late during a disease and may even be required for the viral infection of host cells (30). To

investigate the effects of these two genes on IBV proliferation in DF-1, two stable DF-1s expressed the *SLBP* and *P2RX7* genes. Following infection, there was the detection of changes in the viral structural gene. The results showed that *P2RX7* could significantly reduce the replication of viral structural genes, and it is speculated that *P2RX7* plays an antiviral function in IBV infection, with its inhibitory effects against viral infection also confirmed in dengue virus, vesicular stomatitis virus, and densonucleosis virus (DNV). These findings provide new ideas for the prevention, control, and treatment of IBV in the future. In DF-1 cells overexpressing SLBP, there was a little change in the structural genes except for the increased expression of the M gene, and no fully assembled viral particles were observed by electron microscopy, hence suggesting that *SLBP* may not be the key gene that determines the complete replication and assembly of IBV in DF-1. The reasons for the inability of IBV to complete the life cycle in passaged cells are quite complex and require precise coordination of a large number of genes. There are also some unknown genes that could be involved, and therefore, additional discoveries and studies are needed to solve this problem.

## MATERIALS AND METHODS

### Virus and cells

CK/CH/MY/2020 was isolated from the kidney of a chicken, obtained from a farm in Mianyang, Sichuan Province, China, in 2020, and it was stored in the Key Laboratory of Disease Prevention and Control of the Sichuan Province (14). Primary CEFs were then prepared as follows: the chicken embryos were first removed aseptically on an ultra-clean bench before being placed in phosphate-buffered saline (PBS) containing 1% penicillin–streptomycin. The head, limbs, and internal organs were subsequently removed, with the resulting tissues repeatedly cut with ophthalmic scissors until 1-mm$^3$ pieces were obtained. After washing three times with PBS, 5–10 mL of trypsin was then added for digestion in a 37℃ incubator for 10–20 min, with mixing by inversion performed every 5 min. Once the tissues became fluffy and mist-like, an equal amount of 10% cell culture medium was added to terminate the digestion prior to filtration through a 100-mesh cell sieve to obtain a single-cell suspension. The latter was then divided between 9-cm dishes and incubated at 37℃ in a 5% $CO_2$ incubator. The cells were changed after 2 h of cell wall attachment before resuming incubation in 10% cell medium for 24–48 h until the DF-1 and 293T cells reached confluent monolayers.

The DF-1 and CEF cells were cultured at 37℃ with 5% $CO_2$ in minimum essential medium (Gibco, USA) supplemented with 10% fetal bovine serum (Gibco, USA) and 1% penicillin–streptomycin (Gibco, USA) in six-hole plate. When the cells were fused to 80%, a virus solution of 0.5 multiplicity of infection (MOI) was available, which was changed to a 2% Dulbecco's Modified Eagle's Medium complete medium after 2 h. The cells were observed daily for lesions, and virus collection was carried out after the lesions, and three consecutive passages were performed.

### Total RNA extraction and qPCR

Viral RNA was extracted from virus-infected allantoic fluid with TRIzol (Invitrogen, Carlsbad, CA, USA) following the manufacturer's protocol. The total RNA was reversely translated into cDNA using the PrimeScript RT reagent Kit (Takara Bio Inc., Shiga, Japan). The structural genes of CK/CH/MY/2020 were using the following primers (Table 1) and were subjected to qPCR under the following conditions: 40 cycles of 95℃ for 20 s, gene-specific annealing temperature of 60℃ for 30 s, extension for 20 s at 72℃, and a final extension at 72℃ for 20 s. β-Actin was used for the relative quantification.

### IFA and electron microscopy

Viruses were incubated in six-well plates for 24 h before being used for indirect immunofluorescence detection. For this purpose, the cells were fixed with

**TABLE 1** Structural gene qPCR primers

| Name | | Sequence (5′–3′) | Product length |
|---|---|---|---|
| S | F | GTGGTTGTTGTTGTGGATGCT | 112 |
| | R | AGGTCTGTATTGTTCAGTTACCAC | |
| E | F | CTGATGCTTGTTGTTTATTTTGGT | 166 |
| | R | GGTAGACTTTTATTATTCCAACCG | |
| M | F | TGCCGTAGGTTCAATACTCC | 242 |
| | R | GCAAACCTTTTCTTATTTCCGCT | |
| N | F | TGCCGTAGGTTCAATACTCCT | 215 |
| | R | TCACCAGTGTATTTCTGCACC | |
| β-Actin | F | CCCAAAGCCAACAGAGAGAA | 140 |
| | R | CCATCACCAGAGTCCATCAC | |

glutaraldehyde and sent to the Lilly Medical Center for slide preparation and subsequent observation by transmission electron microscopy.

## Transcriptome sample preparation and library construction

The virus was added into six-well plates as described above, and three replicates were set up with equal amounts of cells in cell culture medium as blanks. The uninfected CEF control group was labeled as CEF-1, CEF-2, and CEF-3, while the uninfected DF-1 one was labeled as DF-1, DF-2, and DF-3. Similarly, the CK/CH/MY/2020-infected CEF experimental group was labeled as A-1, A-2, and A-3, while the CK/CH/MY/2020-infected DF-1 experimental one was labeled as B-1, B-2, and /B-3.

The original culture medium was discarded 24 h after inoculation, and after being washed two to three times with PBS, 750 µL of TRIzol was added. Samples were then collected in 1.5-mL RNA-free Eppendorf (EP) tubes and sent to Beijing Novozymes for subsequent processing and sequencing.

RNA was extracted from the collected cell samples by the TRIzol method, with the RNA integrity subsequently checked using an Agilent 2100 bioanalyzer for quality control. Extracted samples of the required quality were then used to build libraries with the NEBNext Ultra RNA Library Prep Kit for Illumina, before evaluating the quality of the resulting libraries with a Qubit 2.0 Fluorometer and an Agilent 2100 bioanalyzer. After passing the quality check, selected libraries were eventually sequenced.

## Verification for the RNA-seq results

Six genes, LYGL, CMPK2, IFI6, OASL, IFIT5, and MX1, which were significantly upregulated by A VS CEF and B VS DF-1 at the same time, were selected, and the qPCR primers were designed and synthesized with reference to the sequences of the corresponding chickens n the National Center for Biotechnology Information (NCBI), and the primer sequences are shown in Table 2. The cDNAs obtained from normal DF-1 cells (DF-1), DF-1 infected with CK/CH/MY/2020 (B), normal CEF cells, and CEF infected with CK/CH/MY/2020 (A) were used as templates, and β-actin was used as an internal reference for qPCR reactions. The RT-qPCR results of six genes in both groups were consistent with the sequencing results, and the difference was highly significant ($P < 0.01$) compared with the control group. Reflecting the correctness of the transcriptome assay results in Fig. 8.

## Transcriptome data analysis

The final sequence data (clean reads) were obtained by detecting sequencing errors as well as the Guanine and Cytidine (GC) content before filtering the raw sequences. Using GRCg7b (registration number GCA_016699485.1) in the NCBI database as the reference genome, the resulting clean reads were precisely compared with the reference genome to obtain information on the position of the reads. Sequences that could not be compared to the reference genome were then assembled and annotated by new transcripts using String Tie software to obtain the predicted new genes.

**TABLE 2** Transcriptome assay results validate primers

| Name | | Sequence (5'–3') | Product length |
|---|---|---|---|
| LYGL | F | CACAGAGAACTGCGAAGCCG | 197 |
| | R | ATCCTTCAGTGCCGTCCCAG | |
| CMPK2 | F | TGGAGACACGAGTTAGTTTCGTT | 236 |
| | R | GATGTCCAGCACCGATCCA | |
| IFI6 | F | GTCTGGTGAGGCAAAATCCT | 160 |
| | R | GGCCTCATTGGACATCATCTG | |
| OASL | F | TGGGAGATGGGGTTGGAGAG | 259 |
| | R | CCTGGTAGCTGGAGAAGCAG | |
| IFIT5 | F | TGGACTTTGCTGAAGGAGGAT | 234 |
| | R | GTAGTTCCCCCAGGTAACGAG | |
| MX1 | F | ACCACCTTCCTTACCAGTACC | 215 |
| | R | TGCAAGGTCGGATCTTTCTGT | |

Quantitative analysis of each gene's expression was performed using the "Counts" feature to obtain the number of reads covering the start to the end of each gene (filtering out non-paired reads with quality values less than 10 and contrasted to multiple regions). The correlation coefficients for intra-group and inter-group samples were then calculated by Pearson's method based on the fragments per kilobase of transcript per million mapped reads values of all genes in each sample. Similarly, correlation heat maps were drawn to reflect the inter-sample correlation, while coexpression Venn diagrams were plotted based on the number of genes repeatedly expressed between samples.

In differential analysis, genes that were differentially expressed between groups were screened and analyzed using the DESeq2 software, with the criteria for significant differences being a |log2FC | of ≥1 and a $P$-adj of ≤0.05. Upregulated and downregulated genes between groups were counted before drawing bar graphs and volcano plots to visualize the results (volcano plot difference criteria |log2FC| = 2). In addition, Wayne plots were also drawn based on the overlap of differentially expressed genes between groups, while heat maps were drawn to show the clustering of differential gene sets.

In enrichment analysis, GO function enrichment analysis was performed on the identified differentially expressed genes using the cluster Profiler software to enrich the genes into BPs, cellular components, and MF. Similarly, the KEGG pathway enrichment analysis was performed, and in both cases, a significance threshold of $P$-adj < 0.05 was applied. The 30 most significantly enriched groups were eventually selected and plotted as histograms. In cases where the number of enriched groups was less than 30, all groups were plotted.

Online protein interaction network analysis was performed using the STRING (http://string-db.org) protein interaction database, with *Gallus gallus* selected as the species.

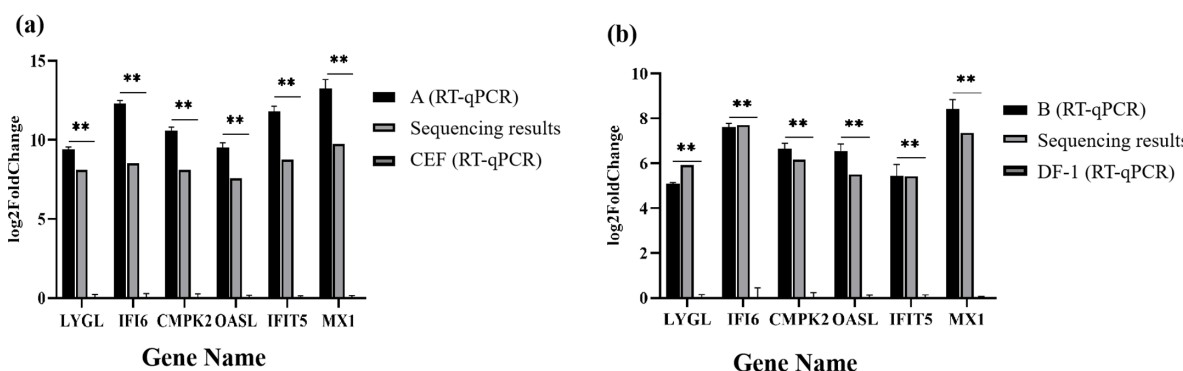

**FIG 8** The sequencing results were verified by RT-qPCR. (a) Difference results of A VS CEF. (b) Difference result of B VS DF-1. *$P$ < 0.05; **$P$ < 0.01.

Each node in the results represents a protein, while differently colored connecting lines between nodes indicate different interactions between proteins.

## Validation of *SLBP* and *P2RX7* gene overexpression

After transcriptome analysis, two genes associated with viral infection were selected and synthesized on the eukaryotic expression plasmid PLVX-PURO with a 6 × His tag at the 5′-end with reference to the sequences in the NCBI database. The two genes were *SLBP* (NCBI accession number: NM_001004403.2) and *P2RX7* (NCBI accession number: XM_001235162.7).

The synthetic plasmid was transfected with pSPAX2 and pMD2.G at 8, 8, and 4 µg in 293T using lipo8000 transfection reagent and packaged as lentivirus. After screening the lentivirus with puromycin, 200 µL of the protein lysate was added to a six-well plate, with the resulting cellular protein samples subjected to the Western blot assay and GAPDH as an internal reference.

## ACKNOWLEDGMENTS

This work was supported by the Earmarked Fund for Modern Agroindustry Technology Research System (CARS-41-K09), Central Government Guided Local Science and Technology.

## AUTHOR AFFILIATION

[1]Key Laboratory of Bio-Resources and Eco-Environment, Ministry of Education, College of Life Science, Sichuan University, Chengdu, China

## AUTHOR ORCIDs

Qingcheng Yang http://orcid.org/0009-0004-4224-5919
Hong-Ning Wang https://orcid.org/0000-0002-7244-9929
Xin Yang http://orcid.org/0000-0002-7940-714X

## AUTHOR CONTRIBUTIONS

Qingcheng Yang, Formal analysis, Writing – original draft, Writing – review and editing | Huiling Gong, Writing – review and editing | Song Liu, Data curation, Formal analysis | Siyu Huang, Formal analysis | Wenjun Yan, Formal analysis, methodology | Kailu Wang, Data curation | Hao Li, Writing – review and editing | Chang-Wei Lei, Writing – review and editing | Hong-Ning Wang, conceptualization, Writing – review and editing | Xin Yang, Funding acquisition, Project administration, Writing – review and editing

## ADDITIONAL FILES

The following material is available online.

Open Peer Review

**PEER REVIEW HISTORY (review-history.pdf).** An accounting of the reviewer comments and feedback.

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
