## [Reviewer comments · Microbiology Spectrum]

Microbiology Spectrum

Differential analysis of IBV-infected primary chicken embryonic fibroblasts and immortalized DF-1

Qingcheng Yang, Huiling Gong, Song liu, Siyu huang, Wenjun Yan, kailu Wang, Hao Li, Chang-Wei Lei, Hong-Ning Wang, and Xin Yang

Corresponding Author(s): Xin Yang, Sichuan University

Review Timeline:

Submission Date:	July 18, 2023
Editorial Decision:	October 9, 2023
Revision Received:	November 13, 2023
Accepted:	November 30, 2023

Editor: Frederick S. Kibenge

Reviewer(s): Disclosure of reviewer identity is with reference to reviewer comments included in decision letter(s). The following individuals involved in review of your submission have agreed to reveal their identity: Bo Hou (Reviewer #1)

Transaction Report:

DOI: <https://doi.org/10.1128/spectrum.02402-23>

October 9, 2023

Dr. Xin Yang
Sichuan University
College of Life Sciences
Key Laboratory of Bio-Resource and Eco-Environment of Ministry of Education, Animal Disease Prevention and Food Safety
Key Laboratory of Sichuan Province
Chengdu, Sichuan 610065
China

Re: Spectrum02402-23 (Differential analysis of IBV-infected primary chicken embryonic fibroblasts and immortalized DF-1)

Dear Dr. Xin Yang:

Link Not Available

Sincerely,

Frederick S. Kibenge

Journals Department
Reviewer comments:

Reviewer #1 (Comments for the Author):

The topic of the manuscript is interesting, which will be useful for the study of IBV through in vitro transmission and pathogenesis. Unfortunately, there are some questions for the author to be published on Microbiology Spectrum.

- 1.RNA-Seq. Huge sequencing datasets or raw data must also be deposited, e.g. as a NCBI BioProject and the accession number provided. Please, read the Information for authors guide for further information.
- 2.The Figures is too blurry and need to be replaced.
- 3.The English proofreading by a native speaker with professional knowledge (especially the discussion part) is required.

Reviewer #2 (Comments for the Author):

IBV is a gamma-coronavirus. Unlike other coronaviruses which could infect different cell lines, while the virus can easily propagate in chicken embryos and primary cell cultures such as CEK cells, adaptation of IBVs on immortalized cell lines is another story. In the manuscript, Yang and colleagues performed transcriptome analysis on IBV-infected CEF and DF-1 cells, and revealed immune-related genes SLBP and P2RX7 can affect replication of the viral structural genes, which provides an interesting point in the research of in vitro pathogenesis of IBVs, since the IBV strain used in the current research can only infect CEF cells. However, the methods part is quite rough. For instance, how was WB performed? Any Ab information? How did IBV in-vitro infection was performed since the timing of virus adsorption and proliferation was quite tricky? Lack of these information has increased the difficulty for the readers to understand the manuscript.

Other minor points required to be addressed are as follows.

1, Fig 1: Arrows pointing to lesions is required.

2, Fig 2: Scale bar is missing.

3, Fig 7: Name of the groups is required to be modified.

4, Line 104, 108: Fig 2 is related to the description of the two paragraphs.

5, Line 208-211: Although these two genes were not included in the key..., they are considered to be important due to their highly significant differential expression as well as ...

6, Any verification for the RNA-seq results?

7, Line 429: References is misspelled.

Staff Comments:

Preparing Revision Guidelines

Please return the manuscript within 60 days; if you cannot complete the modification within this time period, please contact me. If you do not wish to modify the manuscript and prefer to submit it to another journal, please notify me of your decision immediately so that the manuscript may be formally withdrawn from consideration by Microbiology Spectrum.

Dear Editor, Dear reviewers

Thank you for your letter dated October 9. We were pleased to know that our work was rated as potentially acceptable for publication in Journal, subject to adequate revision. We thank the reviewers for the time and effort that they have put into reviewing the previous version of the manuscript. Their suggestions have enabled us to improve our work.

Based on the instructions provided in your letter, we uploaded the file of the revised manuscript. Accordingly, we have uploaded a copy of the original manuscript with all the changes highlighted by using the red typeface. Appended to this letter is our point-by-point response to the comments raised by the reviewers. The comments are reproduced and our responses are given directly afterward in a different color (red). We would like also to thank you for allowing us to resubmit a revised copy of the manuscript.

Reviewer comments:

Reviewer #1 (Comments for the Author):

1.RNA-Seq. Huge sequencing datasets or raw data must also be deposited, e.g. as a NCBI BioProject and the accession number provided. Please, read the Information for authors guide for further information.

The raw data has been uploaded to NCBI under the number SUB13934948.

2. The Figures is too blurry and need to be replaced.

All images in the article have been re-uploaded with clarity guaranteed!

3. The English proofreading by a native speaker with professional knowledge (especially the discussion part) is required.

I have sought out a professional editing agency for English touch-ups.

Thank you for your correction.

Reviewer #2 (Comments for the Author):

1, Fig 1: Arrows pointing to lesions is required.

The lesions have been indicated in the figure by red boxes

2, Fig 2: Scale bar is missing.

After adjusting the clarity of the image the scale is now visible, if it still doesn't work I will re-upload it, thank you for your patience.

3, Fig 7: Name of the groups is required to be modified.

Protein expression of SLBP and P2RX7 and effects on viral structural protein expression. (line 556)

4, Line 104, 108: Fig 2 is related to the description of the two paragraphs.

Relevant comments have been added in lines 102 and 108

I have linked the diagrams to the text in the sections you mentioned, inserting them in the appropriate places. (lines 102 (fig2(a) and lines 108

(fig 2(b)).

5, Line 208-211: Although these two genes were not included in the key..., they are considered to be important due to their highly significant differential expression as well as ...

I apologize for not being able to understand this part of your question. This part of the question refers to the fact that the two genes selected for validation were not the genes with significant differences in fig3, because when reviewing the genes in question, I noticed that they had antiviral-related effects in other studies.

6, Any verification for the RNA-seq results?

I have added a validation process related to RNA-seq results. (line 380-390 and Table 2)

7, Line 429: References is misspelled. line417

The spelling error has been corrected, thank you very much for your careful guidance!

8. How did IBV in-vitro infection was performed since the timing of virus adsorption and proliferation was quite tricky?

In vitro culture of viruses has been added to the methodology. (line 343-349)

Re: Spectrum02402-23R1 (Differential analysis of IBV-infected primary chicken embryonic fibroblasts and immortalized DF-1)

Dear Dr. Xin Yang:

Your manuscript has been accepted, and I am forwarding it to the ASM production staff for publication. Your paper will first be checked to make sure all elements meet the technical requirements. ASM staff will contact you if anything needs to be revised before copyediting and production can begin. Otherwise, you will be notified when your proofs are ready to be viewed.

Sincerely,
Frederick S. Kibenge
Editor
Microbiology Spectrum

Reviewer #1 (Comments for the Author):

No.